# LEAN: Library-Based Adaptation for Continuous, Federated Fine-Tuning

## Abstract

We consider the problem of learning to adapt a foundation model in a federated setting, particularly the most realistic and general setting: 1) When the local data sets are sampled from different distributions but we want to learn a globally adapted model, 2) Where local agents enter and leave the federation asynchronously at each time tick which is beyond the control of the learning algorithm, and 3) Where the goal is continuous adaptation so that after each time tick, the learned adapter generalizes accurately for all participants that have been seen during training. We propose a simple idea called federated *library-based adaptation* (*LEAN*) for exactly this setting. In library-based adaptation, the system maintains a pool, or "library" of so-called "basis pairs." Agents entering the federation check out basis pairs, update them, and check them in. Library-based adaptation is designed to avoid problems with more conventional methods, such as those based on averaging. In particular, we demonstrate *LEAN* outperforms traditional averaging baselines in both communication and computation cost efficiency across a broad range of important settings, including heavy data skewness and high asynchronicity.

## 1 Introduction

In federated learning (FL) (McMahan et al., 2017; Zhang et al., 2023a; Reddi et al., 2021; Xie et al., 2020), a set of participants collaboratively train a joint model without sharing their local data. This paper focuses on the challenging and realistic setting of a "weak federation," a scenario characterized by a lack of coordination and significant heterogeneity. This setting, often called cross-device FL, is defined by three key constraints:

- System Heterogeneity: Participants, who may be mobile or edge devices, have heterogeneous hardware and network connections. They can join or leave the training process at any time. Consequently, algorithms cannot rely on global synchronization steps.
- Communication Bottlenecks: Participants are resource-constrained, with limited memory, compute, and especially network bandwidth. This makes frequent communication, such as uploading and downloading large foundation models, infeasible and necessitates highly communication-efficient methods.
- Statistical Heterogeneity: Each participant's local dataset is generated from a different distribution, meaning the data is non-IID (non-independent and identically distributed) across the network. This can cause significant challenges for model convergence.

A weak federation in particular captures a classic FL case where many mobile phones collectively train a model over locally stored data (e.g., images or text) (Lim et al., 2020; Nguyen et al., 2022a).

We are specifically interested in parameter-efficient fine tuning (PEFT) in this setting (Dettmers et al., 2023; Lester et al., 2021; Liu et al., 2022b; Hu et al., 2022), due to communication bandwidth concerns in a weak federation. The fundamental question we consider in this paper is *how to reconcile local training results into a single global model during synchronization in a weak federation?* The universal answer is typically some variant of averaging, which is used in nearly every recent proposal for PEFT in a federated environment (Sun et al., 2024a; Zhang et al., 2023a; Cho et al., 2023; Sun et al., 2024b). However, as we argue in the paper, averaging is a questionable choice,

especially in a weak federation. The fundamental problem is that an average of a set of locally trained models or adapters—that can look very different in a weak federation due to the inherent heterogeneity—may not be a meaningful global model, and in fact may be quite poor.

In this paper, we consider a fundamentally different approach that eschews averaging. Instead, the federation maintains a set or *library* of *basis pairs*. Basis pairs are pairs of vectors of the form $(\mathbf{b}^{(i)}, \mathbf{a}^{(i)})$. Assume the federation's goal is to fine-tune a matrix $\mathbf{W}$. To take part in this task, a participant *checks out* a randomly selected subset of basis pairs. That is, when the library consists of $n$ basis pairs $\{(\mathbf{b}^{(i)}, \mathbf{a}^{(i)})\}_{i=1}^{n}$, a participant randomly samples an index set $\mathcal{B}$ of size $m < n$ without replacement from $\{1...n\}$ and locally fine-tunes the basis pairs $\{(\mathbf{b}^{(i)}, \mathbf{a}^{(i)})\}_{i \in \mathcal{B}}$ to serve as an adjustment for $\mathbf{W}$. After fine-tuning, these basis pairs are *checked in*; that is, the participant copies its updated version of the set $\{(\mathbf{b}^{(i)}, \mathbf{a}^{(i)})\}_{i \in \mathcal{B}}$ back into the library. The basis pairs in the library are then used during inference. We call this method *Library-basEd AdaptatioN*, or LEAN. In LEAN, basis pairs are never updated via averaging—in fact, they are only updated locally, by each participant then passed around to other participants for further fine-tuning. We find strong evidence this makes LEAN more robust to heterogeneity and asynchronicity, compared to averaging-based approaches.

There are a number of specific contributions of our work, the advantages to the LEAN method:

- LEAN is very simple and should be easy to implement in a real-life system.
- LEAN addresses the identifiability problem in federated PEFT (see Section 3) while still only requiring transmission of basis pairs, in contrast to FLoRA (Wang et al., 2024), which requires broadcasting a full rank matrix during synchronization.
- We show that LEAN naturally applies to a highly asynchronous situation where the server cannot control when each participant chooses to synchronize. Algorithmically, handling asynchronicity requires little or no adjustment in LEAN, and we show experimentally that LEAN shows high accuracy in such difficult cases.
- We show that LEAN is robust to data heterogeneity, where data across participants are dissimilar.
- Our experiments consider a wide range of datasets, with tasks that include fine-tuning of foundation models for vision and text.

## 2 RELATED WORK

Fine-tuning on the a full model is often prohibitively expensive, especially for Large Language Models. PEFT mitigates these costs. Some common approaches include low-rank adaptation (LoRA) (Hu et al., 2022; Dettmers et al., 2023), p-tuning (Liu et al., 2022a), and diff pruning (Guo et al., 2021), mainly in the context of language models. Such methods have also been extended to vision transformer models as well (Shen et al., 2024; Jia et al., 2022; Zhong et al., 2024).

One of the main approaches in FL is direct averaging of trained parameters after communication, known as FedAvg (McMahan et al., 2017). Many variations of this approach has since been developed such as FedProx (Li et al., 2020), MOON (Li et al., 2021b), FedAdam (Reddi et al., 2021), and so on, that modify local training behaviors or add additional server post-processing. Most existing methods for PEFT in a federated environment use some form of averaging to reconcile locally-trained models. FedIT (Zhang et al., 2023a), FedPETuning (Zhang et al., 2023b), and FedPEFT (Sun et al., 2024a) directly average PEFT parameters across clients after some set of local client training. SLoRA (Babakniya et al., 2023) takes a different, meta-learning approach to construct an appropriate initialization of adapter matrices in LoRA. These so-called "primed" LoRA weights are then used in LoRA fine-tuning with the full weights now frozen.

A commonly addressed problem in Federated Fine-Tuning literature is *client rank heterogeneity*, where the client-trained LoRA ranks are of different ranks due to differing client compute power (Cho et al., 2023). While not explicitly considered in this paper, LEAN can be adapted to this case.

We will consider FLoRA (Wang et al., 2024) carefully later in the paper—FLoRA avoids inherent computational errors incurred when naively averaging individual LoRA adapters (Sun et al., 2024b). However, FLoRA uses averaging, and it requires broadcasting a full rank matrix to each participant.

A number of other asynchronous FL algorithms have been proposed. A simple adaptation of FedAvg, FedAsync (Xie et al., 2020), computes a weight based on the staleness of a received update. FedBuff (Nguyen et al., 2022b) maintains a $K$-sized buffer and delaying global updates until $K$ updates are received. A similar method FedFa (Xu et al., 2024) mitigates problems with waiting delays by using a sliding window buffer.

# 3 BACKGROUND: LoRA, FLoRA, AND FEDERATED LEARNING

LoRA is a standard method for low-rank fine-tuning or adaptation. In LoRA, we are given a model with weight matrix $\mathbf{W} \in \mathbb{R}^{b \times a}$, and the goal is to learn a low-rank $\Delta$ matrix such that the fine-tuned model uses $\mathbf{W} + \Delta$, for $\Delta = \mathbf{B}^T \times \mathbf{A}$, with $\mathbf{B} \in \mathbb{R}^{r \times b}$, $\mathbf{A} \in \mathbb{R}^{r \times a}$. In our discussion, for simplicity, we will ignore any additional LoRA scaling factors.

Because rank $r$ can be small, the value of LoRA for FL is clear—it significantly decreases network transfer cost—and its use for FL has been explored (Zhang et al., 2023a; Cho et al., 2023). The most obvious way to apply LoRA in a federation is via some variant of federated averaging (McMahan et al., 2017; Li et al., 2020; 2021b; Wang et al., 2020). That is, the $i$th participant trains low-rank matrix $(\mathbf{B}^{(i)})^T \times \mathbf{A}^{(i)}$ locally, and then when it is time to reconcile $n$ local models, a global $\mathbf{B}$ is computed as $\mathbf{B} = \frac{1}{n} \sum_i \mathbf{B}^{(i)}$; $\mathbf{A}$ is computed similarly.

**The challenge in federated LoRA: identifiability.** However, there is a significant problem with this approach: the fact that $\mathbf{B}^T \times \mathbf{A}$ is invariant to "co-permuting" the rows of $\mathbf{B}$ and $\mathbf{A}$ (that is, if two rows in $\mathbf{B}$ are swapped, and the matching rows in $\mathbf{A}$ are swapped, the product $\mathbf{B}^T \times \mathbf{A}$ remains unchanged). The effect of this is that there is no obvious way to force participants to learn compatible local LoRA adapters. Two participants, even with with identical datasets, can learn perfect local adapters that happen to use different permutations, and so summing/averaging during reconciliation creates global $\mathbf{B}$ and $\mathbf{A}$ matrices that are useless. Effectively, there is an identifiability problem: for two participants $i$ and $j$ who have trained local low-rank adapters, during reconciliation it is difficult to know which rows at participant $i$ correspond with which rows at participant $j$.

In response to this, Wang et al. (2024) proposed FLoRA. Here the idea is to learn each $\mathbf{B}^{(i)}$ and $\mathbf{A}^{(i)}$ locally, and then to during reconciliation to compute a global adapter $\Delta = \sum_i (\mathbf{B}^{(i)})^T \times \mathbf{A}^{(i)}$. Because each local pair are multiplied *before* summing or averaging, identifiability is no longer a problem. The authors of the FLoRA paper call this "stacking." However, $\Delta$ is no longer low rank. Thus, the scheme used in FLoRA is to ship the global $\Delta$ to each participant during synchronization. Each participant sets $\mathbf{W} \leftarrow \mathbf{W} + \Delta$ (as an alternative, a global $\mathbf{W}$ matrix can be maintained and sent to each participant; this is effectively the same algorithm). Each participant uses this new $\mathbf{W}$ to train a new $(\mathbf{B}^{(i)}, \mathbf{A}^{(i)})$ pair, and the process is repeated at the next synchronization step.

An obvious concern about the FLoRA approach is that it needs to send a full $\Delta$ or $\mathbf{W}$ matrix to each participant, and so FLoRA cannot leverage low rank to save communication cost compared to full rank federated training, at least when the updated model is broadcast. Thus, even if $r = 1$, FLoRA still uses more than half as much communication as full federated training.

# 4 A PROBLEM: AVERAGING CAN LEAD TO AN UNSTABLE ALGORITHM

While FLoRA's stacking mechanism addresses the identifiability problem, there is another significant question: Might we actually expect this algorithm to converge, especially in a weak federation?

Let us focus, for a moment, on the problem of data heterogeneity. At each synchronization step, FLoRA is averaging or summing the locally-learned adapters to compute a global $\Delta$. The problem is that *an average of a large set low rank models, learned over heterogeneous data, may not be be useful.* Intuitively, if the low rank models have "all gone in different directions" during local adaptation, the average may simply correspond to the starting point, or worse.

**Classifying birds in an imaginary archipelago.** It is easy to construct a simple, example case that, by design, will result in questionable FLoRA performance. Consider a synthetic dataset which consists of data describing three species of birds. A feature vector $\mathbf{x}$ describing a bird has three features: bird height (in inches), and bird weight (in ounces), and bid wingspan (in inches). Blue birds have a mean (length, weight, wingspan) of $(40, 20, 20)$, red birds have a mean of $(20, 40, 40)$,

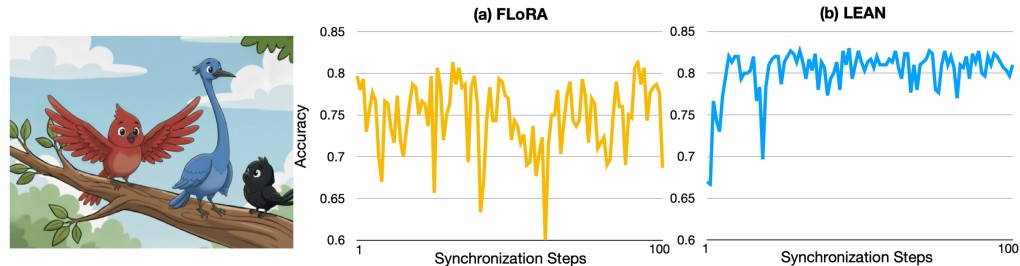

Figure 1: Classification accuracy vs. synchronization steps for a simple synthetic task—learning to classify birds in an imaginary archipelago—with significant class imbalance across participants.

and black birds have a mean of $(5, 5, 5)$. All feature vectors are normally distributed with standard deviation 3.

Now, assume that those birds live on a set of islands in an archipelago. Each island has a monitoring station collecting data on 100 birds. There are three types of islands: type 1 has 88 blue birds, 10 red, and 2 black. Type 2 has 10 blue birds, 88 red, and 2 black birds. Type 3 has 10 blue birds, 2 red, and 88 black. So while it is very easy to identify what type of bird we have from the three features—-the birds are all very different—the various classes are not spread evenly across the islands.

Our goal is to learn to classify birds using a simple model: $f(\mathbf{x}) = \text{softmax}(\mathbf{x} \times \mathbf{W})$. $\mathbf{W}$ was three rows (as there are three features) and three columns (three bird types). Learning is to be done in a federated setting, where at each step, we send the current model to one of each of the three types of islands, and a rank-2 LoRA adapter is trained at each island. The locally-trained adapters are reconciled using FLoRA during syncronization, and the process is repeated.

Objectively, this is an easy learning problem, or at least it would be in a centralized setting. There is significant separation between the classes, but the class imbalance across participants is a problem for averaging, and FLoRA performs erratically. In Figure 1(a), we plot accuracy on a test set that consists of an equal number of each bird type, as a function of training iterations. Accuracy fluctuates significantly over time, occasionally exceeding 80%, but at one point falling to under 60%. This is concerning given the weight matrix is $3 \times 3$ and even a rank-2 adapter should be quite powerful.

## 5 LIBRARY-BASED ADAPTATION

### 5.1 A SIMPLE LIBRARY-BASED APPROACH

If indeed the cause of this behavior is FLoRA's reliance on averaging, it is reasonable to ask: How can we avoid averaging in federated adaptation? One idea is to avoid exchanging a global average, and instead simply exchange low-rank matrices directly. That is, if participant $i$ computes the low-rank pair $(\mathbf{B}^{(i)}, \mathbf{A}^{(i)})$, at a synchronization step it can simply send this pair to another participant, perhaps randomly chosen, for subsequent training. The intuitive argument as to why this might work is that if a participant that has predominantly red birds sends its low-rank matrices directly to a participant with mostly black birds (for example), the receiving participant will further fine-tune the low-rank matrix with the new data, tailoring the model to the presence of the black birds. In our example where we compute three low-rank updates at each synchronization round, we would end up end up circulating three updates throughout training; when it is time to publish the model, we could simply select one of the low-rank updates, or perhaps average all three at the very end.

### 5.2 MAINTAINING A LIBRARY OF BASIS PAIRS

One obvious problem with this idea is that for any low-rank pair $(\mathbf{B}^{(i)}, \mathbf{A}^{(i)})$ to collect information from $n$ different participants, it will have to circulate through at least $n$ different synchronization steps. This means that many synchronization rounds may be necessary to obtain a reasonable model.

We can alleviate this problem by not circulating low-ranking matrices, but by instead circulating *basis pairs*. That is, the federation maintains a set, or a *library* of pairs of vectors of the form

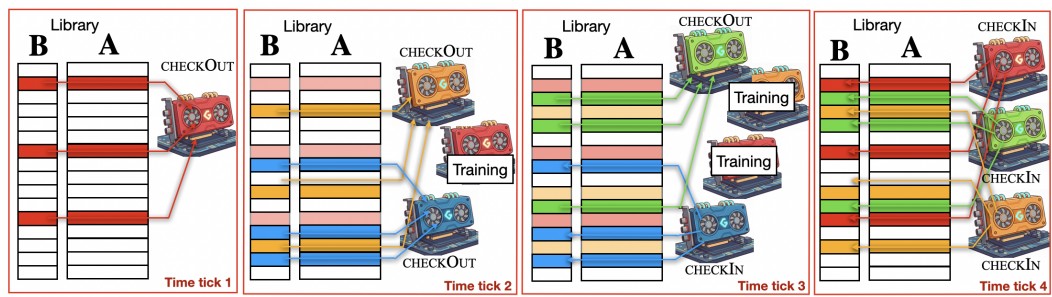

Figure 2: Asynchronous progression of the LEAN algorithm. At time tick 1, a single participant (red) checks out $m = 3$ of $n = 16$ basis pairs. At time tick 2, the red participant is training, and two addition sets of basis pairs are checked out. At time tick 3, the blue participant checks in its pairs, another participant (green) enters, and two participants (red and orange) are still training. At time tick 4, all participants check in the remaining basis pairs.

$(\mathbf{b}^{(i)}, \mathbf{a}^{(i)})$ which we designate the *basis pairs*. Assume the federation's goal is to fine-tune a matrix $\mathbf{W}$. During a synchronization step, a participant *checks out* a randomly selected subset of basis pairs - When the library consists of $n$ basis pairs $\{(\mathbf{b}^{(i)}, \mathbf{a}^{(i)})\}_{i=1}^n$, a participant randomly samples an index set $\mathcal{B}$ of size $m < n$ without replacement from $\{1...n\}$. The participant then locally fine-tunes the basis pairs using weight matrix $\mathbf{W} + \Delta$ with $\mathbf{W}$ fixed and $\Delta = \sum_{i \in \mathcal{B}} (\mathbf{b}^{(i)})^T \times \mathbf{a}^{(i)}$. After fine-tuning, at the next synchronization step these basis pairs are *checked in*; that is, the participant copies its updated version of the set $\{(\mathbf{b}^{(i)}, \mathbf{a}^{(i)})\}_{i \in \mathcal{B}}$ back into the library.

Note that each basis pair that is checked out was likely last trained on a different participant. Thus we might expect much faster adaptation/mixing when circulating basis pairs as opposed to full low-rank matrices $(\mathbf{B}^{(i)}, \mathbf{A}^{(i)})$. When the time comes to publish a model, we can randomly select $n$ pairs from the library to compute the global $\Delta$, or else use $\Delta = \sum_{i=1}^n \frac{m}{n} (\mathbf{b}^{(i)})^T \times \mathbf{a}^{(i)}$ (we use the latter in the experiments in this paper).

We call this method *Library-basEd AdaptatioN*, or LEAN. To show the potential advantage of the LEAN approach, we run the same experiment with blue, red, and black birds, using $m = 2$ for each participant and $n = 6$ basis pairs in the library. As we see in Figure 1(b), the accuracy of the adapted model as a function of the number of synchronization steps is better compared to FLoRA. LEAN is generally more accurate and more stable, even though (a) both methods train rank 2 local adapters, and (b) LEAN has less communication cost, as it never needs to communicate a full rank matrix to a participant.

One thing to note is that LEAN's advantage in this case *is a direct consequence of the skew in class distribution across participants*. Were there no skew, averaging would be far more meaningful and LEAN's advantage may vanish. That said, while this example has focused on statistical or data heterogeneity, we will show that LEAN's avoidance of averaging helps with asynchronicity as well—as we will show in the next section, LEAN's core protocol requires no synchronization steps because there is no aggregation.

## 5.3 THE FULL LEAN ALGORITHM

The full LEAN algorithm is given as Algorithm 1. Note that in the general case, LEAN can work in a fully asynchronous setting, due to the fact that no averaging is required. When a participant is done with local fine tuning, it simply calls CHECKIN on the server to return the fine-tuned basis pairs, and they are copied into the library. When it is ready for more work, a participant calls CHECKOUT on the server to obtain a set of basis pairs to fine-tune. At any time, GETWEIGHTMATRIX can be called to obtain the current, fine-tuned matrix. Asynchronous execution is demonstrated in Figure 2.

A few points regarding Algorithm 1 bear a bit more discussion. First, note that the library is initialized to size $n$, but its size may be increased dynamically during CHECKOUT if there are not enough basis pairs to satisfy client demand. Second, note that during training, a proximal term may be used (if $\lambda > 0$; see line 30) to limit the amount to which a given participant can modify a basis pair. We

have found that this may be useful, especially later during training, and when data across participants are very heterogeneous. When a basis pair is shipped to a "unique" participant, that participant can pull it away from the globally optimal value, and a proximal term can help.

---

**Algorithm 1** LEAN: Federated Learning with a Basis Pair Library

---

1: **Server-side data:** Library size $n$, global basis pair library $\mathbf{B} \in \mathbb{R}^{n \times b}$, $\mathbf{A} \in \mathbb{R}^{n \times a}$, checkout status vector $\mathbf{c}$ of length $n$ initialized to zeros; standard checkout size $m$
2: **Shared data:** Global model weights $\mathbf{W}$

3: **procedure** CHECKOUT  ▷ Server-side execution, called by a participant.
4:     Let $\mathcal{I} \leftarrow \{i | \mathbf{c}_i = 0\}$  ▷ Compute the set of unused bases
5:     **if** $|\mathcal{I}| < m$ **then**  ▷ Check if there are enough unused bases
6:         $n' \leftarrow n + m - |\mathcal{I}|$  ▷ Not enough, so extend the set
7:         Append $(n' - n)$ rows to $\mathbf{B}$; init by copying random existing rows from $\mathbf{B}$
8:         Append $(n' - n)$ rows to $\mathbf{A}$; init by copying random existing rows from $\mathbf{A}$
9:         Append $(n' - n)$ scalar values to $\mathbf{c}$; init to 0
10:         Add $\{n, ..., n'\}$ to $\mathcal{I}$; $n \leftarrow n'$
11:     **end if**
12:     Sample an index set $\mathcal{B}$ of size $m$ without replacement from $\mathcal{I}$
13:     Set $\mathbf{c}_i \leftarrow 1$ forall $i \in \mathcal{B}$  ▷ Remember these are checked out
14:     Let $\mathcal{P} = \{(\mathbf{B}_i, \mathbf{A}_i\}_{i \in \mathcal{B}}$  ▷ We will return the sampled rows from $\mathbf{B}$ and $\mathbf{A}$
15:     **return** $(\mathcal{P}, \mathcal{B})$  ▷ Return basis pairs and their original indices
16: **end procedure**

17: **procedure** CHECKIN$(\mathcal{P}, \mathcal{B})$  ▷ Server-side execution, called by a participant.
18:     **Input:** Updated basis pairs $B$ and their indices $\mathcal{B}$.
19:     **for all** index $i \in \mathcal{B}$ **do**  ▷ Loop through all pairs to check in
20:         Remove a pair $(\mathbf{b}, \mathbf{a})$ from $\mathcal{P}$
21:         $\mathbf{B}_i \leftarrow \mathbf{b}$; $\mathbf{A}_i \leftarrow \mathbf{a}$; $\mathbf{c}_i \leftarrow 0$  ▷ Insert the pair back into the library
22:     **end for**
23: **end procedure**

24: **procedure** ENTERFEDERATION$(\mathcal{D}, m, \lambda)$  ▷ Participant-side execution.
25:     **Input:** Participant's local dataset $\mathcal{D}$, desired basis set size $m$, proximal weight $\lambda$
26:     // *Step 1: Check out basis pairs from the server.*
27:     $(\mathcal{P}, \mathcal{B}) \leftarrow$ server.CHECKOUT$(m)$.
28:     // *Step 2: Locally fine-tune the checked-out basis pairs (LoRA).*
29:     Construct a loss function $\mathcal{L}$ over local data, parameterized on $\mathbf{W} + \sum_{(\mathbf{b}, \mathbf{a}) \in \mathcal{P}} \mathbf{b}^T \times \mathbf{a}$
30:     $\mathcal{P}' \leftarrow$ Train$(\mathcal{L}, \mathcal{P}, \mathcal{D}, \lambda)$  ▷ $\mathbf{W}$ is fixed; proximal penalty $\lambda||\mathcal{P} - \mathcal{P}'||_2$ used during training
31:     // *Step 3: Check in the updated basis pairs to the server.*
32:     server.CHECKIN$(\mathcal{P}', \mathcal{B})$
33: **end procedure**

34: **procedure** GETWEIGHTMATRIX  ▷ Server-side execution; access to current weight matrix.
35:     **return** $\mathbf{W} + \frac{m}{n} \sum_{i=1}^{n} \mathbf{B}_i^T \times \mathbf{A}_i$
36: **end procedure**

---

## 6 EXPERIMENTS

In our experiments, we test the ability of LEAN to fine-tune modern transformers in a federated environment. As our focus is on learning in a weak federation, we will specifically consider experiments examining the effects of data/statistical heterogeneity, asynchronicity, and limited communication bandwidth.

**Tasks considered.** We consider two standard fine-grained image classification datasets/tasks: Oxford VGG 102-Flowers (Flowers) (Nilsback & Zisserman, 2006) and CUB-200-2011 Birds (Birds) (Wah et al., 2011). We also consider CIFAR100 (Krizhevsky, 2009). Finally we consider the Chem-LLMBench (Guo et al., 2023) Reaction Prediction task (ChemRxnPred) for text-based processing.

In the ChemRxnPred task, we are given chemical reactants in SMILES format(Weininger, 1988) and must generate the predicted SMILES format of the reaction outputs.

**Transformers tested.** For image classification, we use a ViT-Huge (Dosovitskiy et al., 2021) model pretrained on ImageNet-21k (Deng et al., 2009). The full-rank transformer blocks have rank $R = 1280$ while our local training LoRA use $r = 128$. There is an additional classification head of the same rank. For QA tasks we use Qwen-3 0.6B (Yang et al., 2025) which has $R = 1024$ blocks. Here we use $r = 64$ local training rank for LoRA.

In a federated setting, it may not be realistic to tune hyper-parameters to a specific FL task. Hence, we do not tune hyperparameters for each task. We choose reasonable parameters and use those parameters for all tasks using the model. Additional details of model hyperparameters are provided in the Appendix.

**Data/statistical heterogeneity.** Dirichlet sampling (Li et al., 2021a;b; Reddi et al., 2021) is a standard method of non-i.i.d. dataset generation. In the image tasks, we use a variant of Dirichlet sampling to partition images across participants, according to their class labels. We use a Dirichlet parameter $\rho = 1$ for "lightly" skewed data (that is, some classes are more prevalent on each participant, but there is not significant skew) and $\rho = 0.01$ to produce heavy skew, where participants have data mostly from one or two classes. In the language task (with no class labels) we partition data uniformly at random.

**Simulating asyncronicity.** Some of our experiments assume stochastic departures and arrivals. Assume there are $N$ participants. At each time tick, a non-participating site becomes a participant with probability $p = \frac{1}{N}$, and performs a CHECKOUT. It then generates a duration to CHECKIN sampled from a Pareto distribution, which is governed by the scale factor $m$ and shape factor $\alpha$. We use a scale factor $m = 25$. $\alpha = 1.16$ produces the "classic" 80-20 Pareto curve which we use as a "light" level of asynchronicity. $\alpha = 0.9$ flattens the curve which results in a difficult case where stragglers are more prevalent; and we designate this the "heavily asynchronous" setting.

**Federated learning methods tested.** *FedAvg with LoRA*: An approach that averages the local LoRA parameters learned (Sun et al., 2024b). *FedProx with LoRA*: The same, but using a proximal penalty term (Li et al., 2020). *FLoRA*: the stacking-based approach discussed in Section 3 (Wang et al., 2024). *LEAN*: as described in this paper. *LEAN w/o Prox*: LEAN with a 0 proximal weight.

One question with LEAN is how to deploy a model for inference. Algorithm 1 suggests using all basis pairs in the library— but does this give the best results? We run a simple experiment– for the Birds dataset, 25 participants, light data heterogeneity, light asynchronicity (as in Section 6.2) we train using LEAN, and sample various numbers of pairs from the library, testing accuracy (Table 1). What we find is that using more basis pairs than the rank of the matrix being fine-tuned (1280 in this case)

| Num. Pairs | Final Acc. (%) |
|---|---|
| 128 | 39.05 |
| 640 | 56.54 |
| 1280 | 59.78 |
| 3200 | 60.46 |

Table 1: Effect of the number of basis pairs used. LEAN, Birds.

produces diminishing returns, and so we cap the number used at the full rank for experiments. However, we emphasize that in standard LoRA deployment (Hu et al., 2022) the adapters are multiplied and added into the frozen weight matrix, and *so for a typical deployment of LEAN we would simply use the full library via standard LoRA inference.*

In order to adapt non-LEAN methods to the asynchronous setting, we use FedFa (Xu et al., 2024) where updates are averaged using a sliding window. In experiments with $N = 10, 25$ participants, we choose window length 5, while for $N = 100$ we use 20.

As a baseline, we will also consider centralized, non-federated fine tuning ("Central").

**Statistics reported.** In experiments where a single "accuracy" number is reported, it is computed as the greatest average test accuracy obtained by three consecutive models during training. We also report the total bytes transferred by the federation during training and FLOPs required to reach high accuracy; "DNC" (or "did not converge") is reported for these quantities if a method did not achieve at least 95% of the accuracy of the best, non-Central method in the experiment.

| Method | Final Acc (%) | Bytes (GB) | FLOPs (Exa) |
|---|---|---|---|
| Central | 99.74[1] | N/A | N/A |
| FedAvg | 12.12 | DNC | DNC |
| FedProx | 17.14 | DNC | DNC |
| FLoRA | 99.25 | 161.45 (2.27×) | 498.88 (1.02×) |
| LEAN w/o Prox | 97.79 | 131.97 (1.86×) | 912.02 (1.86×) |
| LEAN | 98.05 | 71.11 | 491.09 |

Table 2: Comparison of methods in a synchronous setting training on the VGGFlowers dataset (Nilsback & Zisserman, 2006).

| Part. | Dataset | Central Acc. (%) | LEAN Acc. | LEAN Bytes (GB) | LEAN FLOPs (Exa) | FLoRA Acc. | FLoRA Bytes | FLoRA FLOPs |
|---|---|---|---|---|---|---|---|---|
| | Flowers | 99.74[1] | 89.34 | 16.47 | 113.42 | 89.03 | 45.69 (2.77×) | 140.31 (1.24×) |
| 10 | Birds | 89.40[2] | 63.08 | 44.16 | 304.40 | 21.72 | DNC | DNC |
| | CIFAR100 | 94.55[1] | 79.74 | 79.07 | 54.60 | 1.19 | DNC | DNC |
| 25 | Flowers | 99.74[1] | 94.77 | 71.82 | 494.20 (1.18×) | 93.43 | 134.91 | 417.29 |
| | Birds | 89.40[2] | 59.78 | 164.12 | 1132.61 | 7.82 | DNC | DNC |
| 100 | Flowers | 99.74[1] | 74.73 | 268.87 | 1856.38 | 8.80 | DNC | DNC |

Table 3: Comparison of LEAN and FLoRA for various numbers of participants with both light skewness and asynchronicity. We additionally provide Central training accuracy as a reference.

## 6.1 THE SIMPLE SYNCHRONOUS CASE

We begin with a simple case: a fully synchronous setting where all participants enter the federation at the same time, then train locally, and finish local training at the same time (in this setting, the non-LEAN methods do not need to use a queue of recent models to deal with asynchronicity). We use light heterogeneity (Dirichlet parameter $\rho = 1$).

Table 2 gives results for the VGGFlowers data set. In Figure 3 we show the test accuracy of the various methods on this data set as a function of the bytes transferred.

There are a few key findings. First, FedAvg and FedProx do not work well in combination with LoRA—this is not surprising, given the identifiability problem discussed earlier in the paper. As a result, we not consider them further in these experiments. Also, we found that there is a significant advantage to using a proximal penalty in LEAN. The accuracy is not that much lower than LEAN with a proximal term, but the proximal term significantly aides in convergence. Thus we will not consider LEAN w/o Prox further, and only consider LEAN and FLoRA in subsequent experiments. In this simple case, with little statistical heterogeneity and full synchronicity, the final accuracy of FLoRA is very good, surpassing LEAN—though FLoRA does require much more communication to converge, given the fact it must transmit full rank matrices.

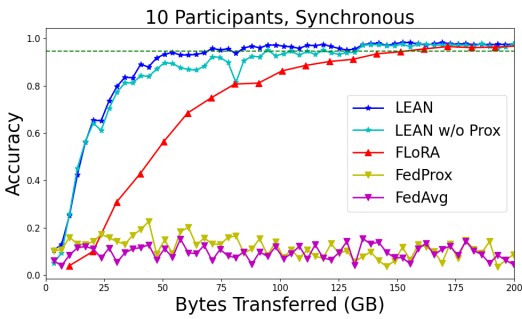

Figure 3: Accuracy as a function of bytes transferred, on 10 participants, synchronous. The dashed line indicates 95% of centralized fine-tuning accuracy.

## 6.2 LIGHT DATA HETEROGENEITY AND ASYNCHRONICITY

We now consider a somewhat more difficult case: participants start and finish training lightly asynchronously. The data are again lightly heterogeneous across participants. Table 3 shows results on

---

[1]Dosovitskiy et al. (2021)

[2]Conde & Turgutlu (2021)

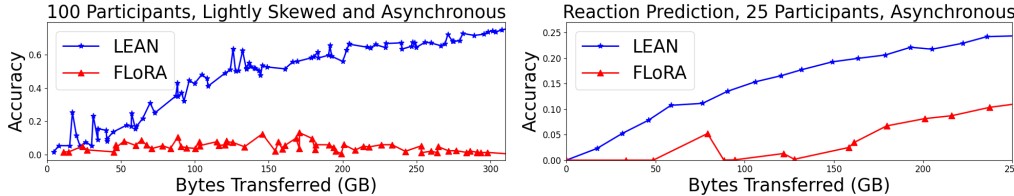

Figure 4: Comparison of LEAN with FLoRA for 100 participants on the Oxford Flowers dataset (left), and 25 participants (right) for the Reaction Prediction task (textual data).

| Part. | Dataset | LEAN | | | FLoRA | | |
|---|---|---|---|---|---|---|---|
| | | Acc. (%) | Bytes (GB) | FLOPs (Exa) | Acc. | Bytes | FLOPs |
| 10 | Flowers | 69.85 | 1269.81 | 183.79 | 3.71 | DNC | DNC |
| | Birds | 14.87 | 21.41 | 147.72 | 0.62 | DNC | DNC |
| | CIFAR100 | 26.53 | 61.61 | 424.44 | 1.16 | DNC | DNC |
| 100 | Flowers | 23.46 | 223.21 | 1540.29 | 3.54 | DNC | DNC |

Table 4: High data heterogeneity, low asynchronicity.

| Part. | Dataset | LEAN | | | FLoRA | | |
|---|---|---|---|---|---|---|---|
| | | Acc. (%) | Bytes (GB) | FLOP (Exa) | Acc. | Bytes | FLOPs |
| 25 | Flowers | 94.26 | 64.12 | 441.98 | 71.30 | DNC | DNC |
| | Birds | 56.52 | 136.90 | 945.13 | 0.92 | DNC | DNC |
| 100 | Flowers | 78.60 | 372.89 | 2575.86 | 12.50 | DNC | DNC |

Table 5: Low data heterogeneity, high asynchronicity.

various image classification tasks for LEAN and FLoRA in this setting. We also plot, in Figure 4, test accuracy as a function of bytes transferred for two specific lightly-asynchronous cases.

Table 3 clearly shows that FLoRA, using a running average to deal with light levels of asynchronicity, is unable to produce reasonable results outside of the Flowers data set with 25 or fewer participants. In all other cases, it produces quite poor accuracy results and the results are typically poorer with more participants. Figure 4 shows that FLoRA generally lags behind LEAN in terms of accuracy as a function of bytes transferred.

### 6.3 The Heavily asynchronous and heterogeneous setting

Next, we consider the most difficult case: high levels or asynchronicity or heterogeneity. In table 4, we show some results in the case with high data heterogeneity (and low asynchronicity). In table 5, we consider the case of heavily asynchronous participation (and high heterogeneity).

Table 4 shows that FLoRA is particularly vulnerable to high levels of data heterogeneity across participants. In the four cases tested, it never exceeds an accuracy of 4%. LEAN is also moderately affected but does much better: Flowers accuracy with 10 participants drops from nearly 90% accuracy (10 sites) to a bit less than 70% accuracy. Interestingly, both methods are more robust to high asynchronicity among the participants compared to levels of data heterogeneity (table 5), but LEAN is again far more robust.

## 7 Conclusion

In this paper, we have considered the problem of low-rank adaptation using a LoRA-like strategy in a federated setting. In particular, we have focused on the challenging problem of a *weak federation*, which is characterized by system heterogeneity, communication bottlenecks, and statistical/data heterogeneity. We have argued that the averaging-based approaches that dominate the scientific literature are perhaps of limited utility using modern models (such as transformers) in the case of a weak federation. We have proposed LEAN, which is a new method for LoRA-like adaptation that does not use averaging.

## 8 REPRODUCIBILITY

Model and federated learning simulation details are provided in Section 6 and Appendix A. Anonymized code specific for this work will be provided in the GitHub repository `https://github.com/leanbibliophile/LEAN`.

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

## A  ADDITIONAL RELATED WORK

### A.1  NON-AVERAGING APPROACHES TO FEDERATED LEARNING

In this work we emphasize non-averaging federated reconciliation as an alternative to averaging algorithms such as FedAvg (McMahan et al., 2017) or FLoRA (Wang et al., 2024). A notable class of federated algorithms is decomposition approaches, such as FjORD (Horváth et al., 2021), Fed-Dropout (Wen et al., 2022), or HeteroFL (Diao et al., 2021), where the a central model is partitioned into smaller sub-networks which are distributed at client sites. These smaller models can help mitigate computation and communication costs compared to the typical FedAvg full-model broadcast as well as mitigate the impact of heterogeneous data (Hu et al., 2023). However, a fundamental drawback of these partitioning methods is their lack of scalability - Because the central model is of fixed size, partitioning to large numbers of clients can fail because the sub-models can become arbitrarily or impossibly small. LEAN avoids this pitfall of previous works by taking instead a bottom-up approach where the sub-models are of fixed-size, as LoRA adapters, and the full-model - the library - is materialized by concatenating and reusing the sent adapters.

### A.2  PRIVACY-PRESERVING FEDERATED LEARNING

While not a focus of this work, privacy and security are major concerns in the broader Federated Learning setting (Bonawitz et al., 2016; Wei et al., 2020). LEAN decomposes and shuffles LoRA adapter updates during the CHECKIN procedure to anonymize individual updates. However because individual rank-1 LoRA pairs are persisted in the library, these individual pairs may be vulnerable to membership or other inference attacks (Nasr et al., 2019). Some works have examined membership inference attack specifically in LoRA (Ran et al., 2025), but on the complete LoRA adapter rather than a decomposed single rank. The extent of leakage in a rank-1 library adapter is beyond the scope of this work.

For mitigating potential privacy leakage, LEAN is compatible with differential privacy (Wei et al., 2020) approaches. We acknowledge LEAN lacks compatibility with secure aggregation (Bonawitz et al., 2016), unlike averaging-based methods, which is a potential vulnerability during the server-side update, i.e. CHECKIN.

## B  MODEL AND FL SETTINGS

### B.1  FEDERATED SIMULATION

All code was run on PyTorch (Paszke et al., 2019) framework with arbitrary random seed 926.

In the federated simulations, *LEAN* models were trained locally for 150 batches before their weights are sent to the server while FLoRA, FedAvg, and FedProx were trained on 200. An exception is for Qwen models, where their local batches are 50 for both LEAN and FLoRA.

Specific algorithmic details in Section 6 for implementing our Dirichlet heterogeneity and asynchronous simulation are provided in this work's associated GitHub repository.

FLOPs computation in simulations was aided by *calflops* Python library (https://pypi.org/project/calflops/0.0.2/)

### B.2  MODELS

The Vision Transformer implementation we used is based on https://github.com/lukemelas/PyTorch-Pretrained-ViT.

Pre-trained model weights for H-14 ViT (https://huggingface.co/google/vit-huge-patch14-224-in21k) and Qwen3-0.6B (https://huggingface.co/Qwen/Qwen3-0.6B) were sourced from Huggingface. A script and configs are provided in our source code to adapt the H-14 weights to the ViT implementation we used.

ViT models were trained with a learning rate of $6 \times 10^{-3}$ and batch size of 8. Qwen models were trained with a learning rate of $3 \times 10^{-4}$ and batch size of 4.

In the case of proximal models (Li et al., 2020), we used a proximal coefficient of $3 \times 10^{-3}$, which we found to be broadly effective.

### B.3 DATASETS

The Oxford VGG 102-Flowers dataset (Nilsback & Zisserman, 2006) was downloaded from the original website: `https://www.robots.ox.ac.uk/~vgg/data/flowers/102/`

CUB-Birds-200-2011 (Wah et al., 2011) was downloaded from the original website: `https://www.vision.caltech.edu/datasets/cub_200_2011/`

CIFAR100 (Krizhevsky, 2009) was provided by the Python *torchvision* library.

We processed all image datasets with a $224 \times 224$ center cropping to fit them to resolution of the the ViT's pre-trained dataset.

ChemLLMBench (Guo et al., 2023) was downloaded from the original GitHub repository: `https://github.com/ChemFoundationModels/ChemLLMBench`. We used the provided USPTO_50k dataset for training and testing our models. Our prompt for the Qwen model was the following:

> You are an expert chemist. Given the reactants SMILES, your task is to predict the main product SMILES using your experienced chemical Reaction Prediction knowledge. Please strictly follow the format, no other information can be provided. You should only reply with SMILES string notations to represent the product. The input contains the reactants and reagents which are split by '.'. The product smiles must be valid and chemically reasonable.
>
> Reactants+Reagents:
> Products:

## C LLM USAGE

We used the Gemini LLM for generating the birds drawing in Figure 1 and the LEAN demonstration diagram in Figure 2. Additionally, Gemini was used for grammar checking, LaTeX assistance, and minor Python debugging.

