# OpenReview forum: "LEAN: Library-Based Adaptation for Continuous, Federated Fine-Tuning"
_ICLR.cc/2026/Conference — Submitted to ICLR 2026_

### Official Review · Reviewer_Pyje · 2025-10-26

**Soundness:** 3
**Presentation:** 1
**Contribution:** 3
**Rating:** 4
**Confidence:** 4

**Summary:**

The paper proposes LEAN (Library-basEd AdaptatioN), a novel framework for weak federated learning that replaces global parameter averaging with a basis-pair library. Each client checks out several basis pairs ((b, a)) from the server, locally fine-tunes them (similar to LoRA), and checks them back in. During inference, the server reconstructs updates by combining multiple basis pairs.

**Strengths:**

Novel and Intuitive Design Concept: The idea of exchanging basis pairs instead of averaging entire low-rank adapters is both elegant and practical. It provides an intuitive explanation of why LEAN might better handle heterogeneous data and asynchronous participation, avoiding the identifiability issues of LoRA averaging.

Clear Focus on Weak Federated Settings: The paper’s motivation is well aligned with realistic federated environments—characterized by asynchronous clients, resource constraints, and highly non-IID data. This problem framing is timely and relevant.

**Weaknesses:**

Writing and Logical Issues: The manuscript contains several grammatical and stylistic errors, “may not be be useful” (double “be”) and “Qwem-3 0.6B” (should be Qwen), which suggests a lack of careful proofreading.
The text states blue birds have a mean (length, weight, wingspan) of (40, 20, 20) and red birds (20, 40, 40), indicating red birds have greater weight and wingspan. However, Figure 1 shows blue birds with all attributes larger than red birds. Could the authors clarify this discrepancy and revise the text or figure for consistency?

Incomplete Baseline Selection: The experiments only compare with FedAvg, FedProx, and FLoRA, omitting several recent and directly relevant baselines such as FedPEFT, FedPETuning, SLoRA, and FedIT.Since these methods explicitly address parameter-efficient or LoRA-based federated learning, their absence substantially weakens the empirical validity of the claimed superiority of LEAN.

Privacy and Security Risks Not Discussed: The basis pairs ((b, a)) stored and transmitted by the server could leak sensitive information about clients’ local data, potentially enabling membership or attribute inference attacks. Unlike aggregated LoRA weights, these basis pairs persist in the central library and are repeatedly reused, which may exacerbate privacy risks.
The paper lacks any discussion or mitigation strategy (e.g., secure aggregation, differential privacy), which is essential for a federated framework.

**Questions:**

Please see the weaknesses above.

---

> ### Author Response · Authors · 2025-11-21
>
> We thank the reviewer for the feedback
>
> Regarding the reviewer’s questions and comments:
>
> **Writing and Logical Issues: The manuscript contains several grammatical and stylistic errors, “may not be be useful” (double “be”) and “Qwem-3 0.6B” (should be Qwen), which suggests a lack of careful proofreading. The text states blue birds have a mean (length, weight, wingspan) of (40, 20, 20) and red birds (20, 40, 40), indicating red birds have greater weight and wingspan. However, Figure 1 shows blue birds with all attributes larger than red birds. Could the authors clarify this discrepancy and revise the text or figure for consistency?**
>
> We have fixed the majority of errors in the writing in an upcoming version of the manuscript and have updated the Figure 1 visual to match the corresponding numbers in the example.
>
> **Incomplete Baseline Selection: The experiments only compare with FedAvg, FedProx, and FLoRA, omitting several recent and directly relevant baselines such as FedPEFT, FedPETuning, SLoRA, and FedIT.Since these methods explicitly address parameter-efficient or LoRA-based federated learning, their absence substantially weakens the empirical validity of the claimed superiority of LEAN.**
>
> We need to clarify that FedPEFT, FedPETuning, and FedIT are all direct applications of FedAvg on PEFT adapters where the adapter weights are averaged and communicated. We have opted to label all of these methods as collectively “FedAvg”. The FLoRA paper (Wang et al. 2024) takes the opposite convention of labeling them as FedIT (Zhang et al. 2024). Regardless of convention, we have addressed these baselines in our tests on FedAvg.
>
> We have added SLoRA (Babakniya et al. 2023) as an additional baseline. Because SLoRA requires an initial full-rank pre-training stage, it is not immediately an asynchronous algorithm and this was the reason for our initial omission. We have chosen to run an initial pre-training stage for SLoRA with participation equal to the size of the FedFa sliding window (5 in the case of 10 and 25 clients) to generate the pre-trained weight used for SVD initialization of the LoRA adapters. Results on 25 clients with heavily asynchronous clients are shown below. Although SLoRA improves over FedAvg and FLoRA, it still trails LEAN by a significant margin. Moreover, we note that SLoRA’s main contribution is the pre-training LoRA intialization that is orthogonal and compatible with LEAN. We will present further results on SLoRA and a more complete analysis in the updated paper.
>
> *25 Clients, VGGFlowers, Heavy Asynchrony*
>
> |Method|Accuracy|Bytes|
> |---|---|---|
> |FLoRA|71.30|DNC|
> |SLoRA|87.23|120.29|
> |LEAN|94.26|64.12|
>
> **Privacy and Security Risks Not Discussed: The basis pairs ((b, a)) stored and transmitted by the server could leak sensitive information about clients’ local data, potentially enabling membership or attribute inference attacks. Unlike aggregated LoRA weights, these basis pairs persist in the central library and are repeatedly reused, which may exacerbate privacy risks. The paper lacks any discussion or mitigation strategy (e.g., secure aggregation, differential privacy), which is essential for a federated framework.**
>
> LEAN is easily compatible with DP noise; local gradient descent would be performed in a differentially private fashion, and by definition the entire learning process would be differentially private.
>
> There is no aggregation in LEAN (except during inference) so secure aggregation is not an option. The downside of LEAN’s lack of aggregation is that aggregation in a large federation can be viewed as protective of privacy without DP noise, which can be lossy. A discussion of these issues can be added to the paper.

---

### Official Review · Reviewer_qKpD · 2025-10-30

**Soundness:** 3
**Presentation:** 2
**Contribution:** 2
**Rating:** 2
**Confidence:** 3

**Summary:**

The paper proposes LEAN, a novel, non-averaging approach for federated fine-tuning in challenging "weak federations" with high asynchronicity and data skew. Instead of aggregating models, clients asynchronously check out, refine, and check in rank-1 "basis pairs" from a central library. This circulation mechanism avoids the instability of averaging-based methods. Strong empirical results on vision and language tasks show LEAN is significantly more robust and communication-efficient than baselines like FLoRA in realistic, heterogeneous settings.

**Strengths:**

1. Clearly identifies and addresses the practical "weak federation" setting, with a compelling critique of standard averaging-based approaches.
2. The "check out/check in" library model is an elegant and creative departure from standard FL, naturally handling asynchronicity and sidestepping common PEFT pitfalls.
3. Provides comprehensive empirical validation on modern models, systematically demonstrating LEAN's superior robustness to data heterogeneity and asynchronicity.

**Weaknesses:**

1. No Theoretical Analysis: The paper is purely empirical, lacking any formal convergence guarantees or theoretical justification for why the method is stable and effective. For a top-tier venue like ICLR, the absence of any theoretical justification is a notable weakness.
2. LEAN introduces several new and important hyperparameters, namely the library size n, the checkout size m, and the proximal weight λ. The paper shows that n has a significant impact on performance (Table 1) but provides no systematic study or guidance on how to set n or m (or their ratio m/n) for a new task. A sensitivity analysis would be crucial for understanding the method's robustness and for enabling practical adoption.
3. The final adapted weight matrix is constructed as W + (m/n) * Σ(all basis pairs). If the library size n is large (e.g., 1280 in the experiments, to match the model's rank), the resulting adapter is a sum of many rank-1 matrices, which is not itself a low-rank matrix. This could introduce significant computational overhead (latency) at inference time compared to a standard single low-rank LoRA adapter..

**Questions:**

1. Can the authors provide any theoretical justification for LEAN's convergence and stability?
2. How sensitive is the method to the choice of library size (n) and checkout size (m), and how should they be set?
3. What is the inference cost of the final summed-pair adapter compared to a standard LoRA adapter?
4. Have the authors considered more intelligent strategies for sampling basis pairs from the library (e.g., based on staleness)?

---

> ### Author Response · Authors · 2025-11-21
> **Part 1**
>
> We thank the reviewer for the feedback
>
> Regarding the reviewer’s questions and comments:
>
> **No Theoretical Analysis: The paper is purely empirical, lacking any formal convergence guarantees or theoretical justification for why the method is stable and effective. For a top-tier venue like ICLR, the absence of any theoretical justification is a notable weakness.**
>
> We definitely respect the reviewer’s point of view here. However, we would argue that what matters is how things perform in practice. We accept that with such an assertion comes an extra responsibility to build an extremely rigorous experimental test suite, and we believe we have done that. In terms of the number and variety of data sets (including both vision and language tasks using modern transformers; we will add experiments on combined vision/language benchmarks as soon as possible) and realistic settings (varying levels of heterogeneity, varying levels of asynchronicity, varying the federation size) we have significantly exceeded the standard level of experimental rigor found in federated learning papers that appear in ICLR and other top-tier venues. In our opinion, the paper has clearly demonstrated the practical utility of LEAN.
>
> **The final adapted weight matrix is constructed as W + (m/n) * Σ(all basis pairs). If the library size n is large (e.g., 1280 in the experiments, to match the model's rank), the resulting adapter is a sum of many rank-1 matrices, which is not itself a low-rank matrix. This could introduce significant computational overhead (latency) at inference time compared to a standard single low-rank LoRA adapter..**
>
> **What is the inference cost of the final summed-pair adapter compared to a standard LoRA adapter?**
>
> For standard LoRA inference (Hu et al. 2022), the frozen full rank weight is replaced by the absorbed combination of the frozen weight and the multiplied adapters. This is the approach we adopt in our implementation of inference in LEAN by absorbing the full library. For most realistic scenarios, except in very small federations, this would be the best option and in such cases there is no dependence between communication and library size/inference pair count.
>
> We believe that Table 1 might give the impression that inference pair count is a hyperparameter and inference costs are a factor to consider. As described above, there is typically no relationship between communication cost/computation cost and the number of pairs used during inference. We will clarify this in the updated paper.
>
> **Have the authors considered more intelligent strategies for sampling basis pairs from the library (e.g., based on staleness)?**
>
> We provide additional ablation runs for different approaches to routing on 25 clients in the heavily asynchronous setting. For similarity-aware routing, we take the cosine similarity between the checkout client’s label distribution and the distribution of the previous client that last checked in the adapter pair, and use a softmax to create a similarity-aware probability distribution that is used to sample basis pairs during checkout. We can also create a similarity-avoiding routing by using the cosine distance instead.
>
> We found little difference in the impact of similarity-favoring routing strategy from random selection in terms of final accuracy, but it converges more slowly as a function of bytes transferred. We found very significant degradation with similarity-avoiding routing.
>
> We provide another study examining the utility of “staleness-aware” routing by taking the time difference between an adapter pair’s last checkin time and the current checkout time, and computing a softmax on these pairs to convert to a probability distribution, prioritizing less stale rows.
>
> There is a modest decrease in maximum accuracy using staleness-aware routing in our most heavily asynchronous setting and a moderate increase in bytes required to converge. Thus it appears that attempting to prevent staleness may be ineffective.
>
> *25 clients, VGGFlowers, Heavy Asynchrony*
>
> |Routing|Accuracy (%)|Bytes (GB)|
> |---|---|---|
> |Random|94.26|71.82|
> |Similarity-favor|94.13|90.85|
> |Similarity-avoid|34.71|DNC|
> |Staleness-aware|92.81|90.92|

---

> ### Author Response · Authors · 2025-11-21
> **Part 2**
>
> **LEAN introduces several new and important hyperparameters, namely the library size $n$, the checkout size $m$, and the proximal weight $\lambda$. The paper shows that $n$ has a significant impact on performance (Table 1) but provides no systematic study or guidance on how to set $n$ or $m$ (or their ratio $m/n$) for a new task. A sensitivity analysis would be crucial for understanding the method's robustness and for enabling practical adoption.**
>
> We want to clarify that library size is not a tunable hyperparameter as library size grows dynamically according to the algorithm described in the paper.
>
> We provide sensitivity analyses for local training rank ($m$) and proximal weight($\lambda$). We will provide a more comprehensive study in the updated paper.
>
> *Local Training Rank*
> *10 Clients, VGGFlowers, moderate heterogeneity and asynchrony*
>
> |Rank|Accuracy (%)|
> |---|---|
> |32|60.80|
> |64|61.88|
> |128|95.37|
> |256|77.43|
>
> On 10 clients, we observe that a $m=128$ checkout rank, as used in our experiments, has the best performance by far, with the fully library matching the full rank ($1280=10\times128$) of the ViT model used. In this small client count case, we observe high sensitivity to the local training rank. However, with a larger population there is a lower sensitivity to checkout rank and we plan to provide these additional results in the updated paper.
>
> *Proximal Term*
> *10 Clients, VGGFlowers, moderate heterogeneity and asynchrony*
>
> |Proximal Term|Accuracy (%)|
> |---|---|
> |0.03|5.11|
> |0.01|91.82|
> |0.007|94.23|
> |0.003|95.37|
> |0.001|13.80|
>
> For the proximal term we find that the range $\lambda\in[0.003,0.01]$ tends to perform well, with the peak at $\lambda=0.003$. However, there is a very significant dropoff outside this range, especially beyond peak point of $0.003$. Further results in the updated paper will address this sensitivity in additional settings.

---

### Official Review · Reviewer_9Ctj · 2025-11-01

**Soundness:** 3
**Presentation:** 3
**Contribution:** 3
**Rating:** 6
**Confidence:** 3

**Summary:**

This paper addresses federated adaptation of foundation models under realistic conditions involving heterogeneous data, asynchronous participation, and continuous adaptation. The authors propose Federated Library-Based Adaptation (LEAN), which maintains a shared library of adaptable basis pairs that agents update collaboratively. Experiments show that LEAN outperforms traditional averaging-based methods in both communication and computation efficiency under diverse and challenging federated learning scenarios.

**Strengths:**

1. Addressing continuous federated learning for foundation models through a library-based adaptation method is a novel and promising direction.

2. The paper is well organized and clearly written, making it easy to follow and understand.

3. The evaluation is comprehensive and effectively demonstrates the proposed method’s effectiveness across diverse federated learning settings.

**Weaknesses:**

1. The contribution of this paper to the federated learning community appears limited.

2. The scalability of the proposed method to larger-scale models (e.g., Qwen3-8B/32B) should be evaluated to make the experimental analysis more complete and convincing.

3. Additional discussion and empirical evaluation on the adaptability of the proposed method to multimodal foundation models would further enhance the contribution and broaden the applicability of this work.

**Questions:**

Please refer to the weaknesses discussed in the previous section.

---

> ### Author Response · Authors · 2025-11-21
>
> We thank the reviewer for the feedback
>
> Regarding the reviewer’s questions and comments:
>
> **The contribution of this paper to the federated learning community appears limited.**
>
> We emphasize our contribution to the Federated Learning field in three areas: 1. Asynchronous FL, 2. Non-averaging FL algorithms, and 3. Federated Fine-Tuning. We will clarify in the updated paper LEAN's relation to prior Federated Learning works.
>
> Asynchrony is a common problem in the *Cross-Device* FL setting (Karimireddy et al. 2021), where problems of weak federations arise due to unreliable client connectivity. Prior works, such as FedAsync (Xie et al. 2019), FedFa (Xu et al. 2024), and FedAT (Chai et al. 2021), have addressed asynchrony by averaging as asynchronous adaptations of FedAvg. However, as we noted in our work the averaging operation can have issues in the presence of highly heterogeneous data.
>
> Some prior works have explored non-averaging schemes in Federated Learning, such as FjORD (Horvath et al. 2021) and HeteroFL (Diao et al. 2021) among others. These methods typically do a top-down, synchronous partitioning of the model, which allows for inherent communication and computational benefits due to broadcasting and training on smaller models. However, these methods scale poorly to large client counts and lack native asynchrony. LEAN addresses both these issues by taking a bottom-up construction of the centrally partitioned model - The parameter library - via LoRA bases.
>
> Finally, it is widely recognized (Babakniya et al. 2023, Cho et al. 2024) that full fine-tuning a foundation model is impractical in a federated setting, particularly with low-compute devices. Our use of LoRA adapters naturally lends itself to this problem of Federated Fine-Tuning.
>
>
> **The scalability of the proposed method to larger-scale models (e.g., Qwen3-8B/32B) should be evaluated to make the experimental analysis more complete and convincing.**
>
> **Additional discussion and empirical evaluation on the adaptability of the proposed method to multimodal foundation models would further enhance the contribution and broaden the applicability of this work.**
>
> We are working on this and hope to provide at least a few experiments on a larger models (such as Qwen3-8B) and a multimodal task (Visual-Question Answering) by the end of the discussion period.

---

### Official Review · Reviewer_hkYf · 2025-11-03

**Soundness:** 2
**Presentation:** 2
**Contribution:** 3
**Rating:** 4
**Confidence:** 4

**Summary:**

The paper proposes LEAN, a library-based alternative to FLoRA for federated LoRA fine-tuning. Instead of stacking per-client adapters, LEAN keeps a shared global pool of rank-1 “atoms” (rows) on the server; clients check out a small set of indices (their rank budget, perhaps), update only those rows, then check in. The server materializes the model by summing all library outer products. The author claims that LEAN avoids row-wise averaging (the identifiability pitfall), supports native asynchrony (no round barriers), and is more robust under weak federations (heavy non-IID + asynchrony) with a proximal regularizer on local updates.

**Strengths:**

- "Identifiability" is an interesting problem framing.
- The framework is straightforward and practical to implement.
- Compared with unbounded stacking, the approach offers more predictable communication cost.
- In several weak-federation regimes, LEAN outperforms—or is more stable than—the stacking baseline.
- Code is released (not yet reviewed in depth on my side).

**Weaknesses:**

- Overclaiming & limited novelty. In my view, LEAN collapses to FLoRA-style stacking whenever indices aren’t reused; so its edge really rides on reuse (for comms), a prox term, and native async. That feels like a modest step beyond FLoRA rather than a substantive new paradigm.

- Motivation not closed. The paper says stacking may falter in weak federations because “many low-rank models go in different directions.” But if LEAN degenerates to stacking without reuse, I don’t see a clear mechanistic reason it would converge better; the argument isn’t theoretically supported.

- Sequential drift worries (my biggest technical concern). With reuse, clients sequentially edit the same row, inviting tug-of-war/negative transfer. There’s no reuse vs. no-reuse ablation, no routing study (random vs. similarity-aware), and no conflict diagnostics (e.g., cosine of successive updates). Also, there’s no theory on stability or convergence of the library dynamics.

- Missing “FLoRA + prox”. Since the proximal term clearly stabilizes LEAN, part of the gain could just be prox, not the library.

- FLoRA doesn’t ship dense full-rank  each round; it stacks LoRA factors, so cost scales with total rank. The paper’s narrative overplays FLoRA’s communication burden.

**Questions:**

1. Compare no-reuse (clone new rows → stacking) vs random reuse vs similarity-aware reuse at matched bytes/rank. Does reuse reduce variance or induce drift?

2. Add these baselines. If FLoRA gets a proximal term and/or runs without barriers, do LEAN’s advantages persist?

3. For reused rows, report cosine similarity between successive client updates, the frequency/extent of “row tug-of-war,” and how the prox weight modulates it.

4. Sweep library size n and inference pair count under fixed communication budgets; where is the sweet spot vs stacking?

5. Does assigning similar clients to the same indices help (or overfit)? Any aging/eviction to prevent stale or monopolized rows?

6.  Can clients use their recently edited rows at inference for personalization? What’s the privacy story for shared atoms (DP noise, secure aggregation, leakage tests)?

7. Larger models and datasets?

---

> ### Author Response · Authors · 2025-11-21
> **Part 1**
>
> We thank the reviewer for the feedback
>
> Regarding the reviewer’s questions and comments:
>
> **FLoRA doesn’t ship dense full-rank each round; it stacks LoRA factors, so cost scales with total rank. The paper’s narrative overplays FLoRA’s communication burden.**
>
> The reviewer is correct that in the FLoRA paper (Wang et al. 2024) the implementation directly ships the stacked rows to the client sites. However, as soon as the number of clients times the size of the adapter exceeds the size of the original weight matrix, this approach has higher communication overhead than simply shipping the full-rank matrix. The authors of FLoRA have acknowledged this in their Discussion section. Shipping the full-rank matrix will be less expensive in many/most realistic scenarios, and it is the case in our experiments. Thus in our experiments we always send the full rank matrix back to all clients.
>
> However, we acknowledge that there are cases when sending the full-rank matrix will be more expensive. We can clarify this in the paper.
>
> **Sweep library size n and inference pair count under fixed communication budgets; where is the sweet spot vs stacking?**
>
> For standard LoRA inference (Hu et al. 2022), the frozen full rank weight is replaced by the absorbed combination of the frozen weight and the multiplied adapters. This is the approach we adopt in our implementation of inference in LEAN. Just as in our FLoRA implementation (see our response above), to perform inference at a client, we would send the new weight matrix to any participant who wanted to perform inference. Again, for most realistic scenarios, except in very small federations, this would be the best option. In such cases there is no dependence between communication and library size/inference pair count.
>
> We believe that Table 1 might give the impression that inference pair count is a hyperparameter and inference costs are a factor to consider. As described above, there is typically no relationship between communication cost/computation cost and the number of pairs used during inference. We will clarify this in the paper.
>
> **In my view, LEAN collapses to FLoRA-style stacking whenever indices aren’t reused; so its edge really rides on reuse (for comms), a prox term, and native async. That feels like a modest step beyond FLoRA rather than a substantive new paradigm.**
>
> **Compare no-reuse (clone new rows → stacking)**
>
> We assert that there is a very fundamental difference between LEAN and FloRA. When a participant enters the federation, LEAN sends only a small portion of the current model (a randomly-selected set of bases) to the participant, where those bases are updated. No participant ever has access to or trains a version of the global model. The global model is “implicit” and is never materialized until it is time to perform inference.
>
> In contrast, when a participant enters a FLoRA federation, it obtains the current, global model, which it updates using LoRA.
>
> As suggested, we ran an ablation study on “no-reuse” where we constantly generate new rows by cloning existing ones, while existing ones are not re-distributed after their initial checkout. We note that this is equivalent to a naive asynchronous implementation of FLoRA where the stack is grown indefinitely without any synchronizations to update the frozen weight. This method performs poorly because it lacks LEAN’s ability to produce basis pairs that have been sent to multiple sites.
>
> *10 Clients, CUB-200-2011, Heavy Heterogeneity*
> |Method|Accuracy|
> |---|---|
> |FLoRA|3.71%|
> |LEANProx|69.85%|
> |No-Reuse LEAN|7.92%|
>
> **random reuse vs similarity-aware reuse**
>
> **Sequential drift worries (my biggest technical concern). With reuse, clients sequentially edit the same row, inviting tug-of-war/negative transfer. There’s no reuse vs. no-reuse ablation, no routing study (random vs. similarity-aware), and no conflict diagnostics (e.g., cosine of successive updates).**
>
> With heterogeneity, a tug-of-war is always possible no matter what method is used, as different participants tend to pull the model in different directions. LEAN attempts to handle heterogeneity by retraining already-trained bases, FLoRA by stacking (effectively, aggregating locally-trained models). LEAN may be susceptible to “forgetting”, but aggregation is problematic as well. It is easy to construct a very simple case (using a two-parameter linear regression) where an average of two locally-trained models is much worse than either locally-trained model.

---

> ### Author Response · Authors · 2025-11-21
> **Part 2**
>
> As suggested, we provide additional ablation runs for different approaches to routing on 25 clients in the heavily asynchronous setting. For similarity-aware routing, we take the cosine similarity between the checkout client’s label distribution and the distribution of the previous client that last checked in the adapter pair, and use a softmax to create a similarity-aware probability distribution that is used to sample basis pairs during checkout. We can also create a similarity-avoiding routing by using the cosine distance instead.
>
> We found little difference in the impact of similarity-favoring routing strategy from random selection in terms of final accuracy, but it converges more slowly as a function of bytes transferred. We found significant degradation with similarity-avoiding routing.
>
> *25 clients, VGGFlowers, Heavy Asynchrony*
>
> |Routing|Accuracy (%)|Bytes (GB)|
> |---|---|---|
> |Random|94.26|64.12|
> |Similarity-favor|94.13|90.85|
> |Similarity-avoid|34.71|DNC|
>
> **For reused rows, report cosine similarity between successive client updates, the frequency/extent of “row tug-of-war,” and how the prox weight modulates it.**
>
> We agree. We are working on providing these results.
>
> **Any aging/eviction to prevent stale or monopolized rows?**
>
> We provide another study examining the utility of “staleness-aware” routing by taking the time difference between an adapter pair’s last checkin time and the current checkout time, and computing a softmax on these pairs to convert to a probability distribution, prioritizing less stale rows.
>
> There is a modest decrease in maximum accuracy using staleness-aware routing in our most heavily asynchronous setting and a moderate increase in bytes required to converge. Thus it appears that attempting to prevent staleness may be ineffective.
>
> *25 clients, VGGFlowers, Heavy Asynchrony*
> |Routing|Accuracy (%)|Bytes (GB)|
> |---|---|---|
> |Random|94.26|64.12|
> |Staleness-aware|92.81|90.92|
>
> **Add these baselines. If FLoRA gets a proximal term and/or runs without barriers, do LEAN’s advantages persist?**
>
> We have run FLoRAprox with in both the 10 client heavily heterogeneous case and the 25 client heavily asynchronous case. We do find that the proximal term can result in a substantial improvement on FLoRA accuracy, but that when the proximal term is chosen to maximize accuracy, FLoRAprox does not seem to converge. For example, we provide the following comparison for the 25 client heavily asynchronous case on the Birds dataset. We take the unsmoothed final accuracy (Hence the small discrepancy from Table 5) and corresponding bytes usage T for LEAN. Then we sample the test accuracy for FLoRAprox at the thresholds T/4, T/2, 0.9T, and T. We find that at 0.9T FLoRAprox actually has higher accuracy than LEAN, but at T it has substantially lower accuracy. We can present more complete results with FLoRAprox in the paper.
>
> *25 clients, CUB-200-2011, Heavy Asynchrony*
> |Threshold Bytes (GB)|LEAN Accuracy (%)|FLoRAprox Accuracy (%)|
> |---|---|---|
> |49.22 (0.25x)|24.68|4.00|
> |98.44 (0.5x)|43.81|18.84|
> |177.192 (0.9x)|52.54|53.11|
> |196.88 (Threshold)|56.89|24.77|
>
> **Can clients use their recently edited rows at inference for personalization?**
>
> Yes. As these have been fine-tuned locally, they will be highly accurate for local use, but will carry global information as well.
>
> **What’s the privacy story for shared atoms (DP noise, secure aggregation, leakage tests)?**
>
> LEAN is easily compatible with DP-SGD; local gradient descent would be performed in a differentially private fashion, and by definition the entire learning process would be differentially private.
>
> There is no aggregation in LEAN (except during inference) so secure aggregation is not an option. The downside of LEAN’s lack of aggregation is that aggregation in a large federation can be viewed as protective of privacy without DP noise, which can be lossy. A discussion of these issues can be added to the paper.
>
> **Larger models and datasets?**
>
> We are working on this and hope to provide at least a few experiments on a larger models (such as Qwen3-8B) and a larger datasets (such as ImageNet) by the end of the discussion period.

---

### Meta-Review · Area_Chair_fRkM · 2026-01-03

**Summary:**

The reviewers have brought about some key weaknesses that remain unaddressed during the rebuttal period. For one, the paper mentions foundation models, but the experiments are narrow with a focus on single tasks/"small" models. Secondly, one of the reviewer mentioned about theoretical justification. It seems the authors could not produce any. Thirdly, it is mentioned that LEAN is compatible with DP noise but no experiments were done or analysis given to validate this claim.

**Reviewer Concerns:**

I do not think the reviewers' concerns were well-addressed. The paper uses the term "foundation models" but the experiments are still very narrow. There is also a lack of theoretical justification or supplemental analysis of DP-FL as requested by the reviewers.

**Reviewer Scores:**

I do  not think the reviewers will change their score even under normal circumstances.

---

### Decision · Program_Chairs · 2026-01-26

Reject